# Knowledge of Child Abuse and Neglect among General Practitioners and Pediatricians in Training: A Survey

**DOI:** 10.3390/children10091429

**Published:** 2023-08-22

**Authors:** Marjolijn Jamaer, Jef Van den Eynde, Bert Aertgeerts, Jaan Toelen

**Affiliations:** 1Faculty of Medicine, KU Leuven, 3000 Leuven, Belgium; marjolijn.jamaer@kuleuven.be (M.J.); jef.vandeneynde@kuleuven.be (J.V.d.E.); 2Department of Cardiovascular Sciences, KU Leuven, 3000 Leuven, Belgium; 3Helen B. Taussig Heart Center, The Johns Hopkins Hospital and School of Medicine, Baltimore, MD 21205, USA; 4Department of Public Health and Primary Care, University Hospitals Leuven, 3000 Leuven, Belgium; bert.aertgeerts@kuleuven.be; 5Leuven Child and Health Institute, KU Leuven, 3000 Leuven, Belgium; 6Department of Development and Regeneration, KU Leuven, 3000 Leuven, Belgium; 7Department of Pediatrics, University Hospitals Leuven, 3000 Leuven, Belgium

**Keywords:** physical signs of child abuse and neglect, physical examination, general practice, pediatrics

## Abstract

Child abuse and neglect is a common, underreported, and worldwide problem. Health care providers play an important role in detecting and reporting this problem. This study examined the knowledge on the clinical signs and mimickers of child abuse among Belgian trainees in family medicine and pediatrics. Participants filled out an anonymous online survey of 15 fictional but realistic cases with either suspicious or non-suspicious signs of abuse or neglect in the context of primary or emergency care. The overall score on all cases, the number of correct answers per case, and the median score were calculated, and the association of the participant’s demographic characteristics with their score was examined using regression analysis. A total of 70 participants completed the survey. The overall median score was 73.3% (IQR 20.0%). The suspicious cases were solved more correctly than the non-suspicious cases (median: 85.7% versus 62.5%, *p* < 0.001). Regression analysis could not reveal a significant association of type and level of education with the performance on the survey. Knowledge of the clinical signs of child maltreatment among pediatricians and general practitioners in training is good, but there is still room for improvement.

## 1. Introduction

Child abuse and neglect is common and occurs in all countries around the world [1,2]. It is a broad phenomenon, which includes not only sexual and physical maltreatment, but also emotional maltreatment as well as neglect [3]. The prevalence rates of sexual and physical abuse during childhood are estimated at around 12.7% and 22.6%, respectively. While data on the other forms of maltreatment are scarce, the estimated prevalence rates for emotional maltreatment, physical neglect, and emotional neglect during childhood are 36.3%, 16.3%, and 18.4%, respectively [2,4].

A total of 6743 dossiers involving minors were reported at the ‘Confidential Centers for Child Abuse’ in Flanders and Brussels in 2019 (pre-pandemic), concerning a total of 8920 minors (about 0.57% of the pediatric population, which is likely an underestimation of the true number of cases) [5]. The largest proportion (51.0%) of notifications was made by schools, childcare facilities, and by health care facilities. In addition, 14.3% of the reports came from welfare organizations and 12.0% from the child’s primary environment, such as the parents or close family [6]. It is clear that health care facilities play a central role in detecting and reporting child maltreatment, including preventive health care providers, family physicians, and school doctors. The organization ‘Child and Family’ focuses on preventive medicine, which includes the detection of possible child maltreatment in children from 0 to 3 years of age [7]. In schools and childcare facilities, the ‘Center for Student Guidance’ has an important role in reporting child maltreatment [6]. This is an institution responsible for student guidance in all recognized schools in Flanders [8]. The coronavirus disease (COVID-19) pandemic has emphasized the problem even more, with a marked increase in cases of child maltreatment in 2020. In Belgium, it was estimated that school closures led to an 80% increase of child maltreatment [9]. The anonymous helpline ‘1712′ reported a 45% increase in calls relating to child maltreatment in 2020 compared to 2019 [10]. Further, the ‘Confidential Centers for Child Abuse’ noticed 4.4% more reports of child maltreatment, and the organization ‘Child Focus’ reported 47% more cases of sexual exploitation [11]. This may be explained by some risk factors of child maltreatment posed by the COVID-19 pandemic and the measures taken to control it, such as reduced monitoring when the caregiver is hospitalized, increased parental stress, anxiety and depressive symptoms, increased job losses, and increased economic stress [12,13,14,15].

However, many cases of child maltreatment go unreported. The global prevalence of child abuse in informant studies (0.3–0.4%) is consistently lower than in self-reported studies (7.6–36.3%), which suggests that although child maltreatment is a common phenomenon, the real extent of the problem is still unknown [2,16]. It can be described as a ‘tip of the iceberg-phenomenon’, where the reported cases constitute the tip, and the unreported cases constitute the invisible submerged part. Within the invisible part, there are two layers, namely, the children whose abuse has been recognized by someone but not (yet) reported to an official agency, and those whose abuse has not yet been recognized at all [2]. To manage this problem, several conditions are necessary: the abuse needs to be recognized, it needs to be reported, and the responsible agencies need to issue a proper response to the abuse. Failure in one of these steps results in missed cases of maltreatment with potentially serious short- and long-term consequences for the child [16,17,18]. Of note, some countries other than Belgium noted fewer reports of child abuse during the COVID-19 pandemic, suggesting that more cases may have been missed [19,20]. The closure of schools and childcare facilities, which normally play an important role in detecting and reporting child maltreatment, may have contributed to this [6,12]. Therefore, to prevent repeated child abuse and its possible consequences, the diligence of the health care providers to detect and act upon these cases is extremely important.

Although the variation of clinical signs and symptoms of child abuse, as well as its “mimickers”, have been extensively described in the medical literature, the knowledge of this problem among clinicians is less well known [21,22,23,24]. Emergency medicine residents and family medicine residents in the United States receive significantly less education about child maltreatment during their training than their pediatric counterparts. The family medicine residents’ knowledge of the evaluation of child maltreatment and their overall comfort of managing a case of child abuse seems to be extremely low [25]. Previous research, however, shows that formal training has a beneficial impact on the necessary competences to recognize and address child abuse. For instance, trainees may develop a better accuracy in distinguishing abusive from non-abusive burn and bruise injuries or a better knowledge and attitude in the management of child abuse cases [26,27,28]. Health care providers who received an education about child abuse were 10 times more likely to report cases of maltreatment than those who did not [27].

Education of health care professionals is the cornerstone to counter the underreporting of child maltreatment as this leads to better competences to recognize it among those who are most likely to encounter it. In particular, the education of general practitioners and preventive physicians is imperative as they constitute the first line of contact with children and their families [17], and a similar argument can be made about the education of pediatricians as they are the second line where children present in a health care context. Therefore, this study’s aim was to examine the knowledge on physical signs of child abuse and neglect among trainees in primary health care and their pediatric counterparts by means of an online survey based on hypothetical case reports.

## 2. Materials and Methods

### 2.1. Study Design

This was a prospective study using an online anonymous survey and was approved by the Research Ethics Committee UZ/KU Leuven (MP016842). Participation was voluntary. The survey included two parts: (1) a questionnaire with demographic questions about the participant and 15 cases and (2) a learning module with feedback on the 15 cases, both administered in Dutch. The demographic data of the participants included their level of education (years 1–5 of residency, junior versus senior), their age, gender, and the number of children they had. The 15 fictional but realistic cases presented in a primary care or emergency setting were the main part of this survey. Each case consisted of an image with the child’s signs or injuries and a contextual story. The participants were asked to indicate how suspicious they thought the case was for child maltreatment (Likert scale: “very suspicious”, “rather suspicious”, “rather not suspicious”, or “not suspicious at all”). At the end of the survey, they received feedback on the cases. The cases and the feedback were written based on the available medical literature and validated by five experts in the field of social pediatrics and child abuse (for a description of the cases, see Table 1). The online survey was sent out via the program Qualtrics. The questionnaire was forwarded to a pilot group in advance to detect any ambiguities and to estimate the time needed to complete the survey. Data collection took place from 30 March 2021 to 5 September 2021. Only the completed questionnaires were used in this research.

### 2.2. Participants

Trainees in primary health care in Flanders were recruited via the online newsletter of the ‘Permanent Educational Committee Family Medicine’. The newsletter was sent via an e-mail and included an invitation with a link to the survey. Trainees in pediatric health care in Flanders received an invitation e-mail via the ‘Belgian Society for Pediatrics’ and the ‘Flemish Society for Pediatrics’. An information letter was attached to both e-mails, clarifying the objective, method, and implications of the study and the possible benefits, risks, and disadvantages of participation. Both in the information letter and at the start of the questionnaire, it was clearly stated that participation was voluntary and that data collection was completely anonymous. Informed consent was asked at the beginning of the survey.

### 2.3. Statistical Analysis

Continuous variables were assessed for normality using the Shapiro–Wilk test and are presented as mean ± standard deviation (SD) or median (interquartile range, IQR), as appropriate. Categorical variables are presented as frequencies and percentages.

The overall total score (i.e., the accuracy of distinguishing abusive from non-abusive cases) was calculated for each respondent based on the number of correct answers. The answers were dichotomously coded as correct or incorrect, whereby “very suspicious” and “rather suspicious” were categorized as “suspicious” and “rather not suspicious” and “not suspicious at all” as “not suspicious”. The correction key was based upon the medical literature and drafted in collaboration with two experts, and indicated for each case whether it was suspicious or not for child maltreatment. The same method was used to calculate the total score on the cases suggestive for child abuse (i.e., the sensitivity of recognizing child abuse) and the total score on the cases not suggestive for child abuse (i.e., the specificity of recognizing child abuse) separately. The median score on the cases suggestive for and the median score on the cases not suggestive for child abuse were compared using the Wilcoxon signed rank test. Additionally, the percentage of correct answers was calculated per case. The distribution of answers per case was visualized by means of a Likert plot. Finally, quadratic regression was performed, assessing whether education, level of education, age, gender, and whether they had children or not, which could predict the overall score of participants. The difference in accuracy between the general practitioners and the pediatricians in training per case was calculated using the Chi-Squared test. All analyses were completed with R Statistical Software (version 4.1.1, Foundation for Statistical Computing, Vienna, Austria) and IBM SPSS Statistics (version 28). The level of statistical significance was set at 0.05.

## 3. Results

### 3.1. Participants

Out of 92 respondents, a total of 70 (45.7% general practitioners in training, 54.3% pediatricians in training) fully completed the questionnaire and were included in the present study. Participant characteristics are presented in Table 2. The majority were women (87.1%) and had no children (92.9%). The median age of the respondents was 26 years (IQR 3, range 24–41 years).

### 3.2. Performance on the Survey

The overall median score for performance on the questionnaire as compared to the correction key was 73.3% (IQR 20.0%, range 33.3–86.7%; Figure 1). In general, the cases suggestive for child abuse were solved better than the cases not suggestive for child abuse (85.7% [IQR 28.6%] versus 62.5% [IQR 25.0%], *p* < 0.001), suggesting that the participants’ sensitivity of detecting child abuse was better than the specificity.

Table 3 shows the number of correct responses per case, while Figure 2 visualizes the responses using a Likert plot. Out of the cases suggestive for child abuse, those with a grid-shaped burn injury (94.3% correct), a hematoma in the shape of a handprint (94.3% correct), a cigarette burn (92.9% correct), and a pinch injury (91.4% correct) were solved best (Table 3, Figure 2). Overall, each case suggestive for child abuse was considered as “suspicious” by the majority of the participants. With regard to the cases not suggestive for child abuse, only the cases about Henoch–Schönlein purpura (97.1% correct) and Nursemaid’s elbow (90.0% correct) were correctly solved by ≥90% of the participants (Table 3, Figure 2). In contrast, non-suggestive cases, such as a pediatric-sized bitemark (30.0% correct), facial petechiae after vomiting (28.6% correct), and lichen sclerosus (10.0% correct), were incorrectly indicated as “suspicious” by the majority of the participants.

On univariable quadratic regression, a significant correlation of gender with overall score was found (male participants scored −13.3% [95% confidence interval, CI −13.3% to −5.2%] worse than female participants, *p* < 0.001), although this significance was not retained in the multivariable regression (−6.7% [95% CI −20.6% to 7.3%], *p* = 0.352, Table 4). Age, type and level of education, and whether the participants had any children did not show any significant correlation with the performance on the survey (Table 4). Only the case about the dermal melanosis was scored better by pediatricians than the general practitioners in training (94.7% correct vs. 68.8% correct, *p* = 0.004).

## 4. Discussion

Overall, signs of child abuse were well-recognized by the participants, although some of the non-suspicious cases were nonetheless classified as either “rather suspicious” or “very suspicious” for child abuse by the majority of participants. Therefore, we conclude that the participant’s sensitivity of recognizing child abuse (classifying the cases suggestive for child abuse as suspicious) was better than their specificity (classifying the cases not suggestive for child abuse as not suspicious). Indeed, using a low detection threshold is a prerequisite to initiate effective management of child abuse or neglect [17]. On the other hand, it is also important to know the clinical signs that may mimic child maltreatment in practice to prevent the trauma and the long-term consequences of false accusation of parents [29]. Our results suggest that the latter should receive more attention in the curriculum of future pediatricians and general practitioners.

An interesting finding regarding the case of a neonate with dermal melanosis is that pediatricians in training answered this question better than the general practitioners in training. Dermal melanosis is an often-described mimicker of child abuse, as the spots may be mistaken for hematomas caused by intentional injury [21,30,31]. The spots are already present at birth or develop within the first weeks and usually fade over the subsequent months or years, thereby being a typical pediatric lesion. They are predominantly seen in Asian and African American infants and typically occur in the lumbosacral region [24,31]. In contrast to hematomas, dermal melanosis does not evolve over a short period of time, such as the time between referral and consultation with the general practitioner [21]. Other features by which melanosis spots can be differentiated from abusive hematomas are, for example, their blurred borders and the absence of associated tenderness, erythema, or swelling [30,31]. Even though there is a low incidence of these spots in the Caucasian (European) population, it is important to know this phenotype in order to distinguish it from child maltreatment [32].

Apart from this case, the type of training did not affect the accuracy by which participants distinguished abusive from non-abusive cases in our study (both a median of 73.3%, *p* = 0.376). This is in contrast to a United States national survey [25], which found that pediatric residents scored significantly better than the general practitioner residents on a knowledge quiz regarding the evaluation of child maltreatment (73.2% versus 58.3%, respectively, *p* < 0.001). This apparent inconsistency might be explained by the fact that proportionately more emphasis is placed on child maltreatment in the training of United States pediatric residents than in the training of their general practice counterparts. Subsequently, Starling et al. concluded that education could play an important role in managing the problem [25]. In Belgium, the amount of time spent on child maltreatment training is approximately the same for residents in both specialties, potentially explaining why both performed equally well in our present study.

A case-based questionnaire proved to be an elegant approach to investigate how adequately medical trainees recognize child maltreatment in clinical practice and distinguish abusive from non-abusive injuries. The cases and the feedback were written based on the available medical literature in cooperation with experts in the field of pediatrics and child maltreatment in order to ensure the validity of the survey. We purposefully included cases with diagnostic ambiguity or in a context of uncertainty, as is often the case in daily clinical practice and allowed a Likert scale-based answer. Yet, for statistical purposes, the correct answers were processed in a binary way (based on expert consensus). The choice was made to use an answer scale with no neutral answer, so that the participants had to choose between ‘suspicious’ or not, which parallels their responsibility in practice to distinguish abusive from non-abusive cases and act accordingly.

The majority of our participants were women, which is in line with the findings of research on online survey response behavior, where among well-educated people, women are generally more likely to participate in an online survey than men [33]. Yet, in our survey, this may also be caused by the higher number of female medical and pediatric trainees. For the education, in theory, the graduate training of all participants should be equivalent, as the different medical schools in Belgium have similar educational criteria. Not only will formal training influence the ability to recognize and address child maltreatment, but also experience, namely, the number of cases of suspected child maltreatment seen in practice [26,27,28]. To investigate the influence of and possible need for additional training on this topic rather than the influence of experience, we decided to take general practitioners and pediatricians in training as a study population rather than experienced general practitioners and pediatricians.

Nevertheless, this study also has some limitations. The questionnaire cannot examine the entire knowledge of a participant on child maltreatment, just as it is open to critique as cases are described in a context of diagnostic uncertainty. Voluntary participation may lead to selection bias, and the absence of face-to-face interaction and direct supervision might affect participants’ engagement and response accuracy. Since there was no opportunity for the participants to indicate that they did not know the answer, bias due to forced choice is possible [34], and as participation was voluntary, self-selection bias may have occurred. In addition, because of the limited number of participants, the results may not be generalizable to an entire cohort of trainees. This survey also did not investigate the previous training of the participants—although it will most likely be identical for most participants because of their basic medical training as students—and the presence of first-hand expertise with the questioned pathologies.

## 5. Conclusions

Based on the results of this study, the knowledge of the clinical signs of child abuse among pediatricians and general practitioners in training is good, but there still seems to be room for improvement, especially for the knowledge about the mimickers of child maltreatment, which were less adequately recognized. Given that previous research shows a beneficial effect of formal training on the necessary competences to recognize and address child abuse, additional education should be part of the postgraduate curriculum of many physicians. These findings imply that it is not only important to know and recognize the abusive signs but also to have knowledge of certain, sometimes rare, conditions that present similarly to child abuse in order to be able to accurately distinguish between abusive and non-abusive cases.

## Figures and Tables

**Figure 1 children-10-01429-f001:**
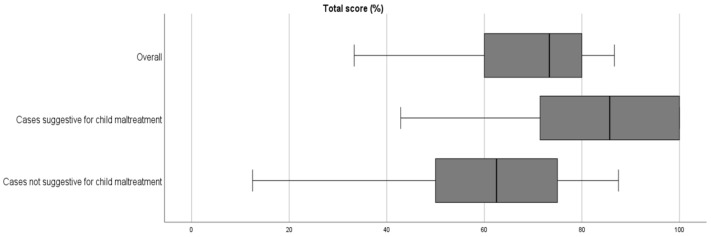
Performance on the questionnaire as compared to the expert score.

**Figure 2 children-10-01429-f002:**
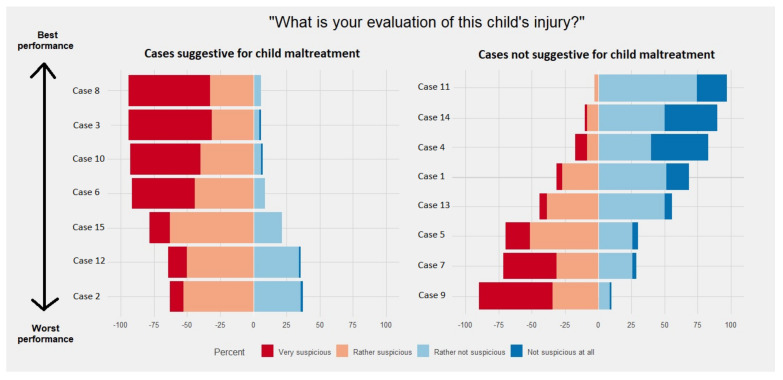
Response to the suspicious and not suspicious cases, cases ranked from best solved to least. Relationship between participant characteristics and performance on the survey.

**Table 1 children-10-01429-t001:** Description of the clinical vignettes and clinical images.

	Vignette	Image Description
Case 1	A mother presents with her 4-year-old daughter in your practice because of a back injury that according to the mother frequently bleeds. It has been there for a few weeks and originated when the girl was ‘with the father’ (divorce with week-to-week arrangement).	The clinical image shows the classical lesion of a pyogenic granuloma: a solitary, red papule above the left scapula on the back of the child. The lesion is not bleeding in the image.
Case 2	A mother comes for a consultation with her 7-month-old daughter because of a rash around the anogenital region. You know that the mother regularly comes and asks for a sickness certificate for her job because of depression. She has become unemployed and is looking for another job. She has four other children, whom you have seen several times at a consultation. You have never seen this child until now. The mother says that this rash was there when she went to change the child’s diaper this morning. According to the mother, the child has had no diarrhea. There is no history of atopy in the family.	The clinical image shows the anogenital region of a female infant with a severe erythematous rash with papulovesicular lesions, fissures, and erosions. It is more severe than the common phenotype of a diaper dermatitis.
Case 3	A grandmother comes with her 13-month-old grandson. The boy has only just started walking a little bit. She is a little worried about the lesion on the ear. She looks after the boy 3 days a week and noticed the injury this morning. According to the mother, the injury has been there since yesterday, but she did not know how it happened. The injury does not go away when pressed on and seems to hurt a little. The child has no fever, there is a normal pulse, and the rest of the clinical examination—except for some eczema spots—is normal.	The clinical image shows a second degree burn in a grid-like pattern on the left earlobe of a toddler.
Case 4	A mother is referred to you by the midwife because of spots on the lower back of a neonate. According to the mother, this has been present for some time and she does not remember any trauma. During the clinical examination, you notice nothing abnormal. The child is of Asian ethnicity.	The clinical image shows the lower back region of a newborn infant with several blueish to blue-grey nummular spots, typical of dermal melanosis.
Case 5	A father comes for a consultation with his 23-month-old son. He is worried because of the injury on his face. The child goes to day-care during the day and they told him that he was bitten by another child. The lesion is sensitive when palpated and the boy, who keeps hiding behind the father, is a bit angry and keeps looking at you suspiciously.	The clinical image shows a toddler with a small bite wound on the left cheek.
Case 6	A couple comes for a consultation with their 3-week-old baby. They are worried because the feeding is not going well, with the baby regularly regurgitatin a lot of milk. According to the parents, the baby seems to have some cramps and always cries in the evening. During the clinical examination, you notice this lesion. Otherwise, there are no eczema lesions and the baby has normal parameters. When you ask the parents how the baby got this lesion, they tell you that they found her this morning with her little leg caught between the bars of the bed.	The clinical image shows both lower legs of an infant with linear reddish lesions in the conformation of a negative imprint of adult digits on the right lower leg.
Case 7	A mother comes in a panic to the consultation with her 18-month-old daughter. Her daughter has just fallen and vomited several times. She fell on the side of her head, in the living room next to the carpet. Immediately after she fell, she started crying. She was never unconscious. You can see these lesions in the face. The rest of the clinical examination is normal.	The clinical image shows the front and side of the face of a toddler with petechiae around the eyes and on the cheeks.
Case 8	A grandmother presents with her 2-year-old grandson because of vomiting and diarrhea for one day. She is worried that he might be dehydrated. At the clinical examination, she sees this non-erasable rash. According to the grandmother, her grandson is a ‘wild child’ who falls and runs into things all the time.	The clinical image shows the left thigh region of a toddler with hand-shaped bruises.
Case 9	A 6-year-old girl comes for a consultation with her mother. The parents are divorced and the girl has just spent a week with her father. The mother is worried because of lesions around the girl’s vagina. The lesions are not painful, and the girl cannot tell much about this abnormality or when it occurred. On further clinical examination, nothing abnormal is observed.	The clinical image shows demarcated white skin lesions, which have an hourglass shape. The skin on the labia majora is atrophic, smooth, and shiny.
Case 10	A mother presents at your consultation with her 6-year-old son because of arunning nose and coughing. During the clinical examination, you notice this injury on the forearm of the boy. The mother does not know when this occurred, and is seeing it for the first time.	The clinical image shows the forearm of a child with a circular second degree burn (crust on the edges of the lesion, diameter 1 cm).
Case 11	A 5-year-old boy comes for a consultation with his parents because of a sudden rash on his limbs. This rash has been present since yesterday, and the parents have the impression it is increasing. The boy has some abdominal pain and had a cold the week before.	The clinical image shows the lower legs of a child with several purpuric lesions (around the ankles and the dorsum of the foot).
Case 12	A father comes with his 3-year-old daughter because of these bruises. The girl is a ‘wild child’ who loves to play and often jumps on the trampoline at home. She has recently started going to school. When asked, the girl had one nosebleed a few weeks ago, but never any gum bleeding. On clinical examination, there are no bruises to be seen anywhere else.	The clinical image shows the upper leg of a child with several linear bruises (four in total, parallel in distribution).
Case 13	A father comes with his son because of red spots in his eye. This morning, the boy woke up with it. You ask the boy how he is doing otherwise (how he feels). He lifts his shoulders and looks at his father. His father then tells you that the boy has been coughing a lot for the past few days and has a sore throat. When asked, the father also says that there have been no nosebleeds or gum bleeding as far as he can remember. On clinical examination, there are no other signs of bleeding and no other lesions. He has no fever.	The clinical image shows the eyes of a child with bilateral conjunctival hemorrhages.
Case 14	A father comes with his 3-year-old daughter. The girl had gone for a walk this morning with her grandfather and the dog when the father received a telephone call from the grandfather saying that the girl suddenly had great pain in her arm. According to the grandfather, she was holding his hand and wanted to jump, whereby the grandfather pulled her arm to prevent her from falling. The girl supports her arm in slight flexion as she walks into the consultation room. She refuses to do anything with her arm. On clinical examination, there is no swelling. She does not allow passive movements. Mild supination of the forearm hurts. The rest of the clinical examination is normal. She has never had anything similar.	The clinical image shows a child on a hospital stretcher with her left arm stretched out parallel to her side, the right arm is in use.
Case 15	A 14-year-old girl presents to your clinic alone. She shows a lesion on her arm that -according to her stsory- has been caused by her stepfather. You know the girl from a very young age but are aware that her mother recently started a new relationship after being divorced many years ago. According to the girl her mother and her new stepfather often have verbal fights Her stpfather is angry with her due to her adolescent and argumentative behaviour and the stress she causes in the household. According to the girl this man often shouts at her, beats her and now has bitten her.	The clinical images shows a circular bruise on the medial side of the forearm. The lesion has a central oval-shaped patch that consists of normal skin. The peripheral part of the lesion shows teeth marks.

**Table 2 children-10-01429-t002:** Sociodemographic characteristics of the participants.

	Categories	N = 70	%
education	General practitioners in training	32	45.7
	Pediatricians in training	38	54.3
level of education	Junior	31	44.3
	Senior	39	55.7
gender	Male	8	11.4
	Female	61	87.1
	X ^a^	1	1.4
age	<25	7	10.0
	25–26	33	47.1
	27–28	16	22.9
	29–30	10	14.3
	>30	4	5.7
number of children	0	65	92.9
	1	3	4.3
	2	1	1.4
	3	0	0
	4	1	1.4

^a^ refer to non male/non female gender.

**Table 3 children-10-01429-t003:** Response to the suspicious and non-suspicious cases, cases ranked from best solved to least.

			Responses (N = 70)			
			Suspicious		Not Suspicious	
			N	(%)	N	(%)
cases suggestive for child abuse	Case 8	Hematoma in the shape of a handprint	66	(94.3)	4	(5.7)
	Case 3	Burn in the shape of a grid	66	(94.3)	4	(5.7)
	Case 10	Cigarette burn	65	(92.9)	5	(7.1)
	Case 6	Pinch injury	64	(91.4)	6	(8.6)
	Case 15	Adult-sized bitemark on a 14-year-old girl	55	(78.6)	15	(21.4)
	Case 12	Linear-shaped hematomas	45	(64.3)	25	(35.7)
	Case 2	Jacquet’s erythema papulosum posterosivum	44	(62.9)	26	(37.1)
cases not suggestive for child abuse	Case 11	Henoch–Schönlein purpura	2	(2.9)	68	(97.1)
	Case 14	Nursemaid’s elbow	7	(10.0)	63	(90.0)
	Case 4	Dermal melanosis	12	(17.1)	58	(82.9)
	Case 1	Pyogenic granuloma	22	(31.4)	48	(68.6)
	Case 13	Subconjunctival hemorrhages after Valsalva-maneuver	31	(44.3)	39	(55.7)
	Case 5	Pediatric-sized bitemark on a 23-month-old boy	49	(70.0)	21	(30.0)
	Case 7	Vomiting-induced petechiae	50	(71.4)	20	(28.6)
	Case 9	Lichen sclerosus et atrophicus	63	(90.0)	7	(10.0)

**Table 4 children-10-01429-t004:** Relationship between participant characteristics and performance on the survey.

		Univariate Quadratic Regression				Multivariate Quadratic Regression			
Variable		Estimate	95% CI	*p*-Value	Nagelkerke Pseudo-R^2^	Estimate	95% CI	*p*-Value	Nagelkerke Pseudo-R^2^
education	General practitioners in training	Ref.				Ref.			0.222
	Pediatricians in training	0.000	−Inf; 12.972	1.000	0.000	−3.810	−13.231; 5.612	0.431	
level of education	Junior	Ref.				Ref.			
	Senior	6.667	−6.305; 6.667	0.054	0.067	0.952	−9.851; 11.756	0.863	
gender ^a^	Female	Ref.				Ref.			
	Male	−13.333	−13.333; −5.214	<0.001	0.132	−6.667	−20.599; 7.266	0.352	
age	Per year increase	0.556	−1.120; 2.137	0.410	0.038	1.905	−0.943; 4.753	0.195	
children	No	Ref.				Ref.			
	Yes	0.000	−8.640; +Inf	1.000	0.000	−5.238	−17.153; 6.677	0.392	

Univariate and multivariate quadratic regression ^a^ 1 participant with gender “X” was excluded from this analysis.

## Data Availability

All data used in this study will be made available to any researcher upon request to J.T.

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
