# Peer review of "Knowledge of Child Abuse and Neglect among General Practitioners and Pediatricians in Training: A Survey"

_children, 2023, doi:10.3390/children10091429_

Round 1

Reviewer 1 Report

The study investigated the knowledge of clinical signs of child abuse and neglect among trainees in primary health care, including general practitioners and pediatricians. It utilized an online survey with hypothetical case reports to assess the participant's ability to distinguish between abusive and non-abusive injuries. The study found that, overall, trainees demonstrated good recognition of abusive signs, but there was room for improvement, especially in recognizing mimickers of child maltreatment. However, the study had limitations, including a small sample size, self-selection bias, and a limited assessment scope, which may affect the generalizability of the findings. Additional education and training were recommended to enhance trainees' competence in accurately recognizing and addressing child abuse cases. The study, in general, is a valuable study that could contribute to the literature after revisions;

1. The sample size of 70 participants is relatively small, especially considering it includes general practitioners and pediatricians in training. This small sample may only adequately represent part of the population of trainees, making it challenging to draw generalizable conclusions. The authors should provide a power analysis for the adequacy of the sample size in this article. 

2. The study's voluntary participation could lead to self-selection bias, where individuals with a particular interest or knowledge of child maltreatment may be more likely to participate. This bias may skew the results and compromise the study's validity. Besides, the study does not account for potential confounding variables influencing participants' responses, such as prior exposure to child maltreatment cases during training. Authors should mention if this is a part of the medical curriculum in the Dutch medical training system. These should be added as a limitation. 

3. Besides, the study did not include a control group of experienced general practitioners and pediatricians, making it difficult to compare the trainees' performance with that of skilled professionals. A control group would have provided valuable insights into the effectiveness of training programs per se.

4. Conducting the survey online may introduce various biases and limitations. For example, the absence of face-to-face interaction and direct supervision might affect participants' engagement and response accuracy. This limitation should also be addressed.

5. The study's cases were fictional and designed to represent scenarios of diagnostic uncertainty. However, selecting cases may need to accurately reflect the complexity and diversity of real-world child maltreatment cases encountered in clinical practice. The authors should provide the rationale for why they selected these cases. A vignette from these cases would help to understand the language and profile of cases and questions. 

6. Discussion is generally satisfactory and needs improvement as mentioned above.

Author Response

Reply to the reviewer’s comments

The study investigated the knowledge of clinical signs of child abuse and neglect among trainees in primary health care, including general practitioners and pediatricians. It utilized an online survey with hypothetical case reports to assess the participant's ability to distinguish between abusive and non-abusive injuries. The study found that, overall, trainees demonstrated good recognition of abusive signs, but there was room for improvement, especially in recognizing mimickers of child maltreatment. However, the study had limitations, including a small sample size, self-selection bias, and a limited assessment scope, which may affect the generalizability of the findings. Additional education and training were recommended to enhance trainees' competence in accurately recognizing and addressing child abuse cases. The study, in general, is a valuable study that could contribute to the literature after revisions.

We would like to thank you for your critical assessment and valuable comments.

  1. The sample size of 70 participants is relatively small, especially considering it includes general practitioners and pediatricians in training. This small sample may only adequately represent part of the population of trainees, making it challenging to draw generalizable conclusions. The authors should provide a power analysis for the adequacy of the sample size in this article. 

You are right that a power calculation is an appropriate prerequisite in modern research. However in order to do so we would need a set significance level (available), statistical power (available), and effect size (unavailable). We do not have an effect size for this survey as there are no published (or unpublished) data on this specific survey available. We can guarantee the internal validity of the survey as it has been reviewed and edited by experts in the field.

  1. The study's voluntary participation could lead to self-selection bias, where individuals with a particular interest or knowledge of child maltreatment may be more likely to participate. This bias may skew the results and compromise the study's validity. Besides, the study does not account for potential confounding variables influencing participants' responses, such as prior exposure to child maltreatment cases during training. Authors should mention if this is a part of the medical curriculum in the Dutch medical training system. These should be added as a limitation. 

This comment is indeed correct, but it is valid for nearly every survey that is published. Voluntary participation is a key element in scientific methodology and is a prerequisite in the ethical approval of any study in the academic field. We have however added this to the limitations of the study. We did mention the limited amount of teaching on child abuse and neglect in the Belgian medical training system in the discussion. We also expounded on the influence of training and exposure in the discussion, but if you feel that this is not adequately phrased (or if additional stresses are necessary), we will gladly adapt the manuscript to your comments.

  1. Besides, the study did not include a control group of experienced general practitioners and pediatricians, making it difficult to compare the trainees' performance with that of skilled professionals. A control group would have provided valuable insights into the effectiveness of training programs per se.

As mentioned in the discussion and by you experienced clinicians are a very relevant group. You are correct that this would be an interesting comparison, yet the context of the study was to asses the knowledge and capacity to diagnose child abuse by young physicians (fresh out of training), as this reflects the quality of the curriculum. Skilled physicians would gain their expertise via the exposure in daily practice, self-study and postgraduate training opportunities. However -regarding the latter- there is at present no high quality CME available for GPs or paediatricians in Belgium.

  1. Conducting the survey online may introduce various biases and limitations. For example, the absence of face-to-face interaction and direct supervision might affect participants' engagement and response accuracy. This limitation should also be addressed.

We have added this into the manuscript.

  1. The study's cases were fictional and designed to represent scenarios of diagnostic uncertainty. However, selecting cases may need to accurately reflect the complexity and diversity of real-world child maltreatment cases encountered in clinical practice. The authors should provide the rationale for why they selected these cases. A vignette from these cases would help to understand the language and profile of cases and questions. 

We have an English version of the survey available for the reviewer, yet the inclusion of this into the supplementary data of this manuscript, may lead to copyright infringement. We have selected images that are readily available from online sources, but for a closed survey there were no copyright problems at stake. For an online publication, this will prove problematic.

  1. Discussion is generally satisfactory and needs improvement as mentioned above.

We have changed the manuscript in accordance with your comments.

Reviewer 2 Report

Dear Authors,

Congratulations on your extensive work, concerning Knowledge of child abuse and neglect among general practi-2 tioners and pediatricians in training: a survey.

Introduction

Could you directly specify study aims?

M&M

The authors mention (line 105) about a learning module with feedback on the 15 cases,  administered in Dutch. Could the authors elaborate more about this study measure?

Perhaps could it be attached (in English) as supplementary material ?

Discussion:

The last part of discussion should be dvided into subsections: strenghts and limitations, future research implications, practical clinical implications, and then study conclusions.

Author Response

We would like to thank the reviewer for providing a critical appraisal of our study and manuscript. 

We have added the study aim into the introduction.

We can provide the reviewer with an Enlish version of the survey, but we may run into issues with copyright infringment if we would add this to the supplemenatry data. Even though many of the images have been accessed via online sources, many may be subject to copyright.

The learning module that was mentioned in the manuscript involved a case-by-case revision of the survey with expert or evidence based explanations on the clinical image or the clinical vignette that was provided.

The specific subsectioning of the discussion is not in the author guidelines of the journal. We can attest that all these mentioned factors are present in the discussion.
